# Influence of the EndoActivator Irrigation System on Dentinal Tubule Penetration of a Novel Tricalcium Silicate-Based Sealer

**DOI:** 10.3390/dj6030045

**Published:** 2018-09-03

**Authors:** Roula El Hachem, Guy Le Brun, Bernard Le Jeune, Fabrice Pellen, Issam Khalil, Marie Abboud

**Affiliations:** 1Department of Endodontics, Faculty of Dentistry, Saint Joseph University, P. O. Box 11-5076 Riad el-Solh, Beirut 1107 2180, Lebanon; issam.khalil@usj.edu.lb; 2Laboratoire OPTIMAG, University of Bretagne Occidentale, 6 Avenue Le Gorgeu, C.S. 93837, 29238 Brest cedex 3, France; guy.lebrun@univ-brest.fr (G.L.B.); bernard.lejeune@univ-brest.fr (B.L.J.); fpellen@univ-brest.fr (F.P.); 3Physics Department, UR TVA, Faculty of Science, Saint Joseph University, B.P. 11-514 Riad el-Solh, Beirut 1107 2050, Lebanon; marie.abboud@usj.edu.lb

**Keywords:** Tricalcium silicate sealer, EndoActivator^®^, dentinal tubules penetration, gap, interfacial adaptation, confocal laser scanning microscope

## Abstract

This study compared the effects of a conventional endodontic needle with an agitation system on a novel tricalcium silicate-based sealer (NTS) in terms of dentinal tubule penetration and interfacial adaptation to a root canal. Fifty single-rooted, recently-extracted human maxillary central incisors were randomly distributed into two homogeneous groups characterized by two different final cleansing systems: Conventional endodontic needle, or EndoActivator^®^. After instrumentation, all the teeth were filled with the gutta-percha single cone technique in conjunction with the novel tricalcium silicate-based sealer. Teeth were horizontally sectioned at 1 and 5 mm from the apex and were observed under a confocal laser scanning microscope (CLSM) at five magnifications. The maximum, mean, and the circumferential percentage of the sealer penetration inside the tubules were measured. Moreover, the gap width was evaluated using Image J software (National Institutes of Health, Bethesda, MD). EndoActivator^®^ did not result in a significantly higher circumferential percentage of sealer penetration than conventional irrigation (*p* > 0.05). However, the gap width was significantly lower with EndoActivator^®^, compared to conventional needles at both 1 mm (*p* = 0.035) and 5 mm (*p* = 0.038). The EndoActivator^®^ irrigation system did not significantly improve the NTS penetration, as compared to the conventional endodontic needle irrigation. Activation of the irrigation reduced the gap width significantly.

## 1. Introduction

The aim of root canal treatment is to eradicate microorganisms, particularly in the apical region, and to prevent bacterial contamination or regrowth. This is usually obtained by the combination of chemomechanical disinfection of the root canal system, followed by three-dimensional obturation [1]. Endodontic irrigants, such as sodium hypochlorite (NaOCl) and ethylenediaminetetraacetic acid (EDTA), have usually been used during endodontic therapy. They dissolve pulp tissue and act on the inorganic portion of dentin, allowing greater canal disinfection and a better obturation of the prepared root canal [2,3]. However, conventional needle irrigation may not allow irrigants to profoundly diffuse, disinfect, and remove debris from the dentinal tubules [4,5]. Therefore, to improve the spreading and efficiency of the irrigant solution, many irrigation devices have been developed, such as Self-Adjusting File^®^, EndoActivator^®^, Irrisafe^®^, and Endovac^®^. EndoActivator^®^ (Dentsply, Tulsa Dental Specialties, Tulsa, OK, USA), a sonically driven irrigation system, activates irrigant solutions using frequencies in the range of 2–3 kHz. The machine accomplishes hydrodynamic activation of the irrigants that is capable of cleaning the root canal system [6] and irregularities, such as lateral canals [7]. After cleaning, canal obturation can be performed by achieving high adaptability of the filling materials [8]. It is crucial to use sealer during the obturation technique in order to minimize voids between the filling material and the canal wall, and to seal dentinal tubules. Newly-developed calcium silicate-based sealers are reported to penetrate tubules to a comparable extent to resin-based sealers, such as AH Plus [9]. These sealers have high pH values during the setting process, induce bioactivity on the material surface, and show a chemical bond to the radicular dentin [10]. In addition, calcium silicate-based sealers show low cytotoxicity, antibacterial activity, and high biocompatibility. These sealers have the potential to induce an osteogenic response [11,12]. A novel tricalcium silicate (NTS)-based sealer composed of tricalcium silicate, tantalum, and calcium oxide has recently been developed [13].

In different studies, the effects of irrigation activation on smear layer removal [14,15], debridement ability [16,17], bacterial eradication [18,19,20], and postoperative pain [21] have been studied. The use of activation devices has been shown to reduce the formation of voids in the filling material and to increase the interfacial adaptation between the sealer and canal walls [22]. It significantly improved tubule penetration of the sealer at 1 to 3 mm from the apex when compared to conventional endodontic needle irrigation [23]. Sonic and ultrasonic activation have been reported to be responsible for the increased penetration of epoxy-amine resin-based root canal sealer into simulated lateral canals [24]. However, the effects of the EndoActivator^®^ irrigation system on bioceramic sealer penetration and interfacial adaptation to root canals have not been studied.

The aim of this study was to compare the effect of the EndoActivator^®^ irrigation system and conventional endodontic needle irrigation on NTS penetration into dentinal tubules and interfacial adaptation to canal walls using confocal laser scanning microscopy (CLSM). The null hypothesis tested was that there were no significant differences in the depth of sealer penetration or in the gap width among the two experimental groups.

## 2. Materials and Methods 

This study was an in vitro study and was approved by the ethic committee of Saint-Joseph University (USJ-2016-61) on 22 September 2016. 

### 2.1. Selection of Teeth

Fifty single-rooted human maxillary incisors with fully formed apices were sterilized and stored at 37 °C and at 100% humidity for two weeks before use. Teeth were obtained from patients whose age varied from 50 to 60 years old. Digital periapical radiographs (SOPIX², ACTEON Group, La Ciotat, France) were taken from the bucco-lingual and mesiodistal directions to confirm the presence of a single canal. The crowns of all teeth were removed close to the cementoenamel junction to get roots with a standardized length of 16 mm. A standard access preparation was made. The working length (WL) was determined by measuring the penetration of a size 10 K-file (Dentsply Maillefer, Ballaigues, Switzerland) introduced passively until it reached the apical foramen and then subtracting 0.5 mm. Samples were distributed into two experimental groups:

Group CN: 25 teeth irrigated with the conventional endodontic needle.

Group EA: 25 teeth irrigated with the EndoActivator^®^. 

Root canals were instrumented to a size #15 K file (Dentsply Maillefer) followed by the ProTaper Universal (Dentsply Maillefer, Ballaigues, Switzerland) rotary instruments to a size of F4 (40/06) in rotary motion. The handpiece was used with an electric engine (X-smart, Densply Maillefer, Ballaigues, Switzerland). The instruments were used according to the manufacturer’s instructions.

In the CN group, each canal was irrigated with a 1 mL of 5.25% NaOCl (Niclor, Ogna, Muggio, Italy) after each instrument. A syringe with a 30-gauge side-vented needle (Max-i-Probe; Dentsply Rinn, Elgin, IL, USA) was used and placed before the binding point with a distance of 2 mm from the WL. Finally, all canals were rinsed successively with 1 mL of 5.25% NaOCl, 1 mL of 17% EDTA and 1 mL of 5.25% NaOCl. Each irrigant was left in place for 30 s. This final irrigation was performed with a needle placed at 2 mm from the WL. This was followed by a final flush with 10 mL deionized water.

In Group EA, the irrigation protocol was the same as in Group CN, but the final irrigation was performed by using the 25/04 noncutting polymer tip of the EndoActivator^®^, placed 2 mm from the WL for 30 s for each irrigant solution. This was followed by a final flush with 10 mL of deionized water.

### 2.2. Root Canal Obturation

All canals were dried with paper points and filled by the single cone technique. NTS was prepared according to the manufacturer’s instructions [13]. To allow analysis under the CLSM, sealers were labeled with Rhodamine B (Sigma-Aldrich, St. Louis, MO, USA) to an approximate concentration of 0.1% in order to provide the fluorescence and to enable confocal laser microscopy assessment. The sealer was placed into the canal, 1 mm short of the working length using a size 30 lentulo spiral (Dentsply Maillefer, Ballaigues, Switzerland). Lentulo was used in light pecking motions within the canal for at least 5 s. A single gutta-percha cone (ProTaper Universal F4, Dentsply Maillefer) was then slightly coated with sealer and placed in the root canal to the WL. The cone was seared off at the orifice level and lightly condensed with a plugger. The coronal opening was filled with a temporary filling material (Cavit, 3M; ESPE, St. Paul, MN, USA) and the samples were stored at 100% humidity and at 37 °C for two weeks to completely set. 

### 2.3. Sectioning of Roots and Preparation of Root Surfaces

After two weeks, the roots were embedded centrally and vertically in orthodontic resin (Dentsply Caulk, Milford, DE, USA). Specimens were stored at 100% humidity at room temperature for the remainder of the study (6–7 weeks). Each specimen was horizontally sectioned using a diamond disk (Buehler, Lake Bluff, IL, USA) with a slow speed (25,000 rpm) hand-piece 1 and 5 mm from the root apex. Samples were then mounted onto glass slides and the coronal surface was polished using sandpapers of 500, 700, and 1200 grit under running water to eliminate the dentin debris produced during root canal shaping and to produce a good reflective surface. The samples submitted to confocal laser microscopy were 2 mm thick.

### 2.4. Confocal Laser Scanning Microscopic Analysis of the Roots

Root canal segments were examined with a Zeiss confocal laser scanning microscope (Carl Zeiss, LSM 780, Jena, Germany) at five magnifications and set in fluorescent mode (at a wavelength of 514 nm), as shown in Figure 1. Digital images were imported into Image J software (National Institutes of Health, Bethesda, MD, USA). To determine the circumferential percentage of sealer penetration, we first outlined and measured the circumference of the root canal wall with the software measuring tool (Figure 2). Next, we outlined and measured the canal walls into which the sealer penetrated the dentinal tubules with any distance. Finally, to calculate the circumferential percentage of sealer penetration, we divided the outlined distances by the canal circumference [25]. Sealer penetration depths into the dentinal tubules were measured at the maximum depth and at four circumferential points (12, 3, 6, and 9 o’clock) for each specimen [26] (Figure 3). Additionally, the width of each gap was measured and pooled for each specimen for comparison [27] (Figure 4). The operator who made the measurements was blinded as to which samples were matched to which group (CN or EA), and the measurements were repeated twice to ensure reproducibility.

### 2.5. Statistical Analysis

The statistical software SPSS for Windows (Version 22.0, Chicago, IL, USA) was used for statistical analysis of the data. The level of significance was set at 0.05. The normality distribution of continuous variables was assessed with Kolmogorov–Smirnov tests. The percentage of penetration, maximum and average depth penetration of the sealer into the dentinal tubules, and the gap width were compared between the two groups (CN and EA) at 1 and 5 mm from the apex. A two-way analysis of variance was conducted, followed by a univariate analysis and multiple comparisons tests of Tukey (HSD). The Mann–Whitney tests were conducted to compare the gap width between different groups.

## 3. Results

### 3.1. Sealer Penetration

#### 3.1.1. Maximum Depth Penetration of Sealer in Dentinal Tubules

The maximum depth penetration of the sealer in dentinal tubules was not significantly different between the two groups at 1 mm (*p* = 0.991) and 5 mm from the apex (*p* = 0.081). The maximum depth penetration was significantly higher at 5 mm compared to 1 mm from the apex for CN (*p* < 0.001) and EA (*p* < 0.001) (Table 1).

#### 3.1.2. Mean Penetration Depth of Sealer in Dentinal Tubules

The mean penetration depth of sealer was not significantly different between the two groups at the 1 mm level (*p* = 0.732). However, it was significantly elevated at 5 mm for the CN group (*p* = 0.016). Moreover, the mean penetration depth was significantly elevated at 5 mm compared to 1 mm for CN (*p* < 0.001) and EA (*p* < 0.001), as shown in Table 1.

#### 3.1.3. Circumferential Percentage of Sealer Penetration into Dentinal Tubules

The circumferential percentage of sealer penetration was not significantly different between the two groups at 1 mm (*p* = 0.633) and 5 mm (*p* = 0.070). However, it was significantly elevated at 1 mm compared to 5 mm for CN (*p* = 0.011) but not significant for EA (*p* = 0.925), as shown in Table 1.

### 3.2. Gap Width

The gap width was significantly lower with EA compared to CN at 1 mm (*p* = 0.035) and 5 mm (*p* = 0.038). The gap width was not significantly different between 5 mm and 1 mm for CN (*p* = 0.660) and EA (*p* = 0.885), as shown in Table 2.

## 4. Discussion

The aim of this study was to compare the effects of the EndoActivator^®^ irrigation system and conventional endodontic needle irrigation on NTS penetration into dentinal tubules and interfacial adaptation to root canal using CLSM. To our knowledge, the effects of the EndoActivator^®^ irrigation system on bioceramic sealer penetration and interfacial adaptation to root canal have not been studied.

Scanning electron microscopy (SEM) [28,29], light microscopy [30], and CLSM have been used to evaluate sealer penetration into dentinal tubules [31,32]. In the present study, CLSM was used to analyze sealer penetration and the gap width as it does not, in contrast to SEM, promote specimen dehydration and produces fewer artifacts than the conventional methods [33]. With light microscopy, it is impossible to differentiate the sealer from the dentin [34]. However, when using CLSM, specimens can be visualized in different depths [8,35]. In a recent study, Tedesco et al. [36] found that the use of CLSM allowed a better evaluation of the depth and quantity of sealer. The sealer was labeled with Rhodamine B since it has no effect on its physical properties [37] and it identifies the sealer within the dentinal tubules [38].

This study did not show a significant difference between the two groups when comparing the mean or maximum penetration depth, or the circumferential percentage of NTS penetration. These results are similar to those reported by other studies [31,39] that showed that sonic activation of irrigants did not significantly improve sealer penetration with respect to conventional irrigation. This may be due to the fact that EndoActivator^®^ did not increase the removal of smear layer as compared with conventional needles [40]. Conversely, Oliveira et al. [41] found that EndoActivator^®^ yields higher results than the endodontic needle in terms of the circumferential percentage of sealer penetration. One reason for this may be the fact that, in their study, each irrigant was used for one minute instead of 30 s and was placed into the canal 1 mm short of the working length instead of the 2 mm adopted in our study. Aksel et al. [42] also showed that irrigant activation may positively impact the quality of the sealer penetration that is achieved with root canal filling. In their study, which compared sealer penetration after irrigation activation, Turkel et al. [43] found that BC sealer exhibited a significantly higher circumferential percentage of sealer penetration than AH Plus. These findings are in accordance with other studies that showed that irrigation activation improves sealer penetration at the middle and apical levels compared to conventional endodontic needle irrigation [23,44,45]. On the other hand, sealer penetration cannot be presumed to be a total index of the absence of smear layer, because the presence of the smear layer may limit, but not totally inhibit, sealer penetration into tubules [32]. Therefore, penetration of the sealer into dentinal tubules might be affected by its physicochemical properties rather than by the activation of irrigants [46]. In this study, the novel tricalcium silicate-based sealer was applied with gutta-percha in a single cone technique due to its wide use in the evaluation of bioceramic-based sealers [47].

The results of the present study showed that the maximum depth of the sealer penetration was better in the coronal thirds than in the apical thirds of root canals in both experimental groups. This can be explained by the fact that the dentinal tubules in the coronal third were present in a greater quantity with larger diameters than those in the apical area [32]. Another possible reason of higher NTS penetration at the coronal thirds can be attributed to better irrigant effects in these regions and more smear layer elimination. The present results confirmed the findings of previous studies that showed significantly superior sealer penetration at greater distances (5 mm) from the apex [31,48]. This may be due to the fact that the apical region has fewer dentinal tubules with smaller diameters, more sclerotic dentin, and more difficult access for irrigants [49], which may lead to less removal of the smear layer in apical thirds when compared to the coronal thirds of the canals.

In our study, fewer gaps were observed at both the 1 and 5 mm levels with EndoActivator^®^ compared to the endodontic needle. In fact, the gap width was significantly lower in Group EA compared to CN at 1 mm (*p* = 0.035) and 5 mm (*p* = 0.038). On the other hand, no significant difference was found between 5 mm and 1 mm in the same group (*p* = 0.660 and *p* = 0.885, respectively). One of the most important requirements of root canal sealer is its adaptation to root dentin. The degree of adaptation depends on a multitude of interacting factors, including the dentin intermolecular surface energy and cleanliness, and the sealer surface tension and its wetting ability [50]. Clinically, the critical root filling area that is susceptible to bacterial leakage is located at the interface between the sealer and dentin [51]. According to Estrela et al. [52], regions containing gaps may increase the possibility of bacterial invasion and recolonization, as bacteria can pass into the interface between the sealer and root canal wall [53]. The purpose of root canal irrigation is to disorganize biofilms produced by bacteria, act on the organic content, and to eliminate the debris induced by root canal shaping [54]. One consequence of root canal irrigation is the change to the interaction of dentine with root canal sealer by affecting the characteristics of the dentine substrate [55]. The most commonly used irrigating agents are NaOCl and EDTA. The proteolytic nature of NaOCl causes some damage to dentine collagen, and this becomes magnified when a sequence of NaOCl–EDTA–NaOCl is used [56,57,58]. The superior adaptation of NTS sealer with irrigation activation in the present study could be due to the dentin cleanliness, and to its physical proprieties and possible changes in the surface energy of the dentine. Additionally, a possible explanation for this could be that this irrigant activation un-obstructs dentinal tubules and exposes dentine collagen fibrils. As a result, sealer penetration into dentinal tubules could be improved [59]. The NTS sealer is composed of tricalcium silicate but includes calcium carbonate in the formulation. The calcium carbonate is added as a filler and acts as a nucleating agent, providing more reaction sites for cement hydration and enhancing the physical properties of the material [60]. The NTS sealer has a lower flow and target to better bond with dentin and to increase bioactivity. It interacts with the physiological solution because phosphorus is drained in greater amounts, thus showing bioactive potential [13]. In a recent systematic review, Neelakantan et al. [61] reported that the improved adaptation to dentine by a bioactive material in the presence of moisture may be due to the process of biomineralization. 

The main finding of this study shows that sonic activation during root canal treatment promotes a lower presence of gaps when bioceramic-based endodontic sealer is used, which may lead to better healing of the periapical infection and may prevent root canal failure.

## 5. Conclusions

Within the limits of this study, irrigant activation did not improve the novel tricalcium silicate-based sealer penetration into the dentinal tubules. However, the interfacial adaptation of the sealer was improved with the EndoActivator^®^, thus reducing the gap region which promotes better root canal treatment outcomes because persisting gaps favor leakage and bacterial contamination, contributing to the failure of endodontic therapy. However, CLSM does not give volume information about gaps—like microcomputed tomography data does [62,63]. Furthermore, future studies with different methodologies and bioceramic-based endodontic sealer may be essential to confirm these results. Whether sonic activation furnishes any significant advantage in ameliorating clinical outcomes remains to be confirmed.

## Figures and Tables

**Figure 1 dentistry-06-00045-f001:**
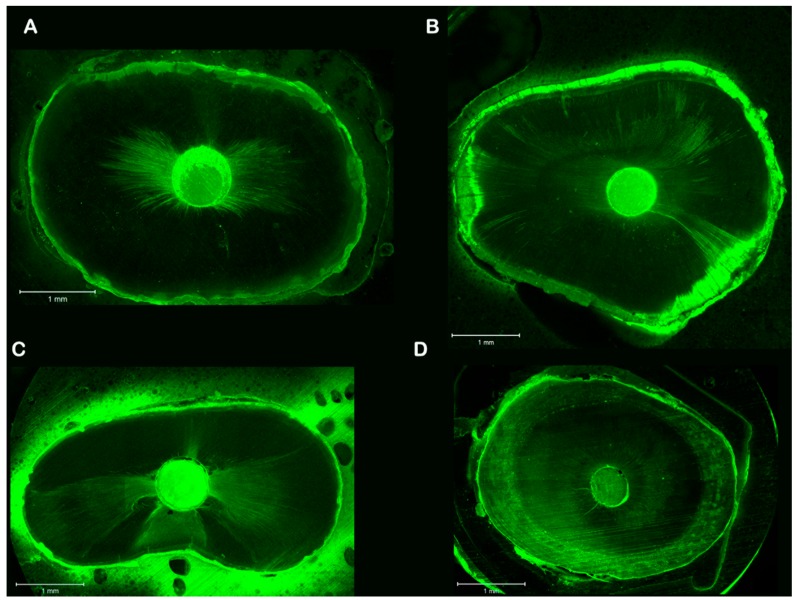
Representative confocal images of sealer penetration on dentinal tubule of the novel tricalcium silicate (NTS) sealer from the two groups at 1 and 5 mm levels. (**A**) EA group at the 5 mm level; (**B**) EA group at the 1 mm level; (**C**) CN group at the 5 mm level; and (**D**) CN group at the 1 mm level.

**Figure 2 dentistry-06-00045-f002:**
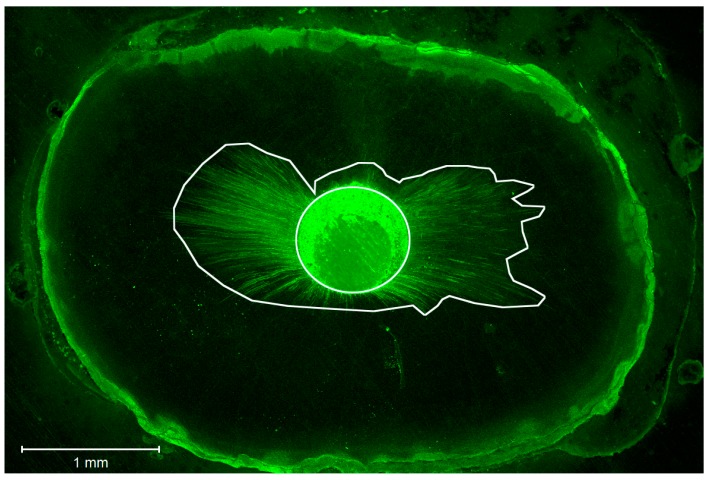
Confocal image showing the measurement of the circumferential percentage of sealer penetration.

**Figure 3 dentistry-06-00045-f003:**
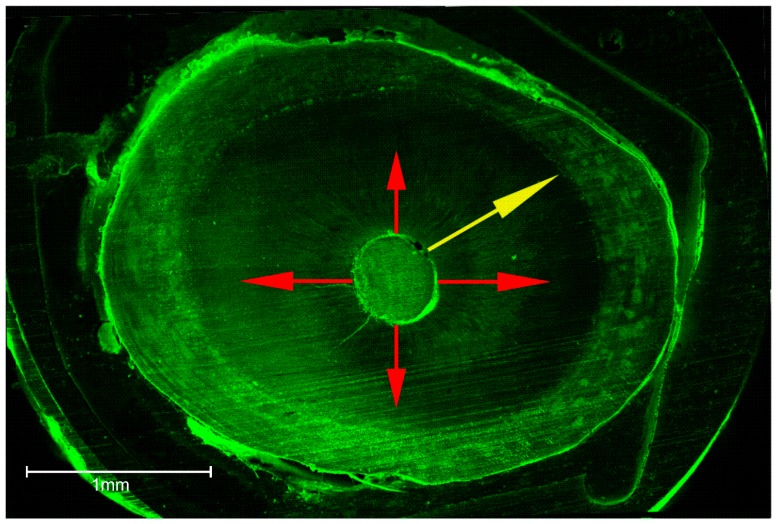
Measurements of maximum depth (yellow arrow) and mean penetration depth at four circumferential depths (red arrows).

**Figure 4 dentistry-06-00045-f004:**
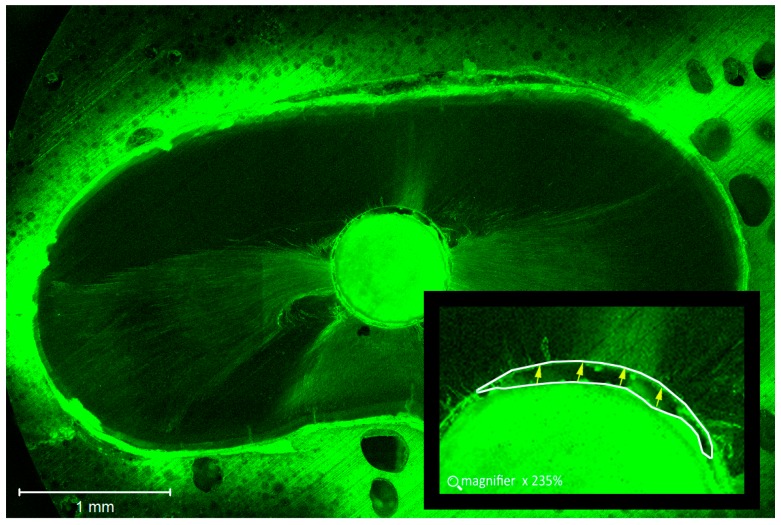
Confocal image showing the gap between NTS sealer and root canal walls.

**Table 1 dentistry-06-00045-t001:** The maximum depth penetration, mean penetration depth, and the circumferential percentage of sealer in dentinal tubules in different groups.

Groups	Level	Maximum Depth Penetration		Mean Penetration Depth		Circumferential Percentage of Sealer Penetration	
		Max (μm)	σ (μm)	Mean (μm)	σ (μm)	%	σ
CN	1 mm	1180.71	±443.47	656.89	±272.94	27.12%	8.51%
	5 mm	1821.97	±338.51	1063.62	±293.53	21.99%	6.58%
EA	1 mm	1181.94	±320.24	630.85	±242.13	26.02%	8.33%
	5 mm	1630.98	±500.83	869.27	±304.60	26.23%	9.52%

**Table 2 dentistry-06-00045-t002:** Mean gap width (µm) in different groups.

Group	Level	Mean (µm)	σ (µm)
CN	1 mm	17.71	±29.24
	5 mm	21.47	±40.08
EA	1 mm	5.21	±19.73
	5 mm	6.35	±25.78

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
