# Peer review of "Influence of the EndoActivator Irrigation System on Dentinal Tubule Penetration of a Novel Tricalcium Silicate-Based Sealer"

_dentistry, 2018, doi:10.3390/dj6030045_

Round 1
Reviewer 1 Report
Congratulation to investigate the Influence of the EndoActivator Irrigation System on Dentinal Tubule Penetration of a Novel Tricalcium Silicate-Based Sealer.
It is necessary to explain in the materials and methods, with detail:
· You used teeth with a single canal?
· How did you made radiography to confirm?
· The Protaper used is PT Universal?
· What was the torque used for each instrument?
· The rotation is in reciprocation or rotary motion?
Author Response
Dear reviewer 1,
We would like to thank you for your valuable comments that we took into consideration. We are very pleased to know that you appreciate our work and recognize its interest and quality.
You can find below a detailed response to all the points.
Best regards,
The authors
Reply to question #1: You used teeth with a single canal?
- Our response: Yes, we used fifty single-rooted human maxillary incisors.
- Revised text in Line 93: Fifty single-rooted human maxillary incisors with fully formed apices were sterilized and stored at 37 °C and 100% humidity for two weeks before use.
Reply to question #2: How did you made radiography to confirm?
- Our response: We used Digital periapical radiographs (SOPIX², ACTEON Group, La Ciotat, France) taken from the bucco-lingual and mesiodistal directions to confirm the presence of a single canal. Similar approach was adopted by Chandra et al. (see Chandra SS, Shankar P, Indira R (2012) Depth of penetration of four resin sealers into radicular dentinal tubules: a confocal microscopic study. J Endod 38:1412-1416. https://doi.org/10.1016/j.joen.2012.05.017)
- Revised text in Line 95-97: Digital periapical radiographs (SOPIX², ACTEON Group, La Ciotat, France) were taken from the bucco-lingual and mesiodistal directions to confirm the presence of a single canal.
Reply to question #3: The Protaper used is PT Universal?
- Our response: We used the ProTaper Universal rotary instruments and we will add this information in the revised text. In our in vitro study we referred to the protocols described in the literature. Many authors who studied sealer penetration have used ProTaper Universal. (see Aksel et al., Generali et al., Akcay et al.)
· Aksel, H.; Küçükkaya Eren, S.; Puralı, N.; Serper, A.; Azim, A.A. Efficacy of different irrigant prorocols and application systems on sealer penetration using a stepwise CLSM analysis. Microsc. Res. Tech. 2017, 80, 1323–1327, doi:10.1002/jemt.22944.
· Generali, L.; Cavani, F.; Serena, V.; Pettenati, C.; Righi, E.; Bertoldi, C. Effect of different irrigation systems on sealer penetration into dentinal tubules. J. Endod. 2017, 43, 652–656, doi:10.1016/j.joen.2016.12.004.
· Akcay, M., Arslan, H., Mese, M., Durmus, N. and Capar, I.D., 2017. Effect of photon-initiated photoacoustic streaming, passive ultrasonic, and sonic irrigation techniques on dentinal tubule penetration of irrigation solution: a confocal microscopic study. Clinical oral investigations, 21(7), pp.2205-2212.
- Revised text in Lines 104-105: Root canals were instrumented to a size #15 K file (Dentsply Maillefer) followed by the ProTaper Universal rotary instruments.
Reply to question #4: What was the torque used for each instrument?
- Our response: We used the handpiece with an electric engine (X-smart, Densply Maillefer, Ballaigues, Switzerland) according to the manufacturers' instructions. (see X-smart® torque card).
- Revised text in Line 107: The handpiece was used with an electric engine (X-smart, Densply Maillefer, Ballaigues, Switzerland) at 250 rpm. The instruments were used according to the manufacturers' instructions.
Reply to question #5: The rotation is in reciprocation or rotary motion?
- Our response: We used the ProTaper Universal rotary instruments in rotary motion and we will mention it in the protocol.
- Revised text in Lines 105-106: Root canals were instrumented to a size #15 K file (Dentsply Maillefer) followed by the ProTaper Universal rotary instruments (Dentsply Maillefer, Ballaigues, Switzerland) to a size of F4 (40/06) in rotary motion.

Reviewer 2 Report
[Suggestions]
Figure 4; A scale bar may be inserted in the magnified figure.
Table 2 and Figure 5; A same data may be presented in the Figure 2 and Figure 5; thus either of these could be deleted from the manuscript.
Author Response
Dear reviewer 2,
We are very pleased to know that you appreciate our work and recognize its interest and quality. We would like to thank you for your valuable suggestions that we took into consideration:
Reply to Suggestion #1: Figure 4; A scale bar may be inserted in the magnified figure.
-Our response: The insert in Figure 4 is a zoom of the confocal image showing the gap region. We added the magnification in the figure 4..
-Revised text in Figure 4 line 197: We added the magnification in the figure 4.
Reply to Suggestion 2: Table 2 and Figure 5; A same data may be presented in the Figure 2 and Figure 5; thus either of these could be deleted from the manuscript
Our response: Since Table 2 and Figure 5 present same data, we removed from the manuscript Figure 5 and its legend (Lines 235).
Revised text in Line 230: we deleted Figure 5 and in line 231 we also deleted Figure 5 and its legend
Best regards,
The authors

Reviewer 3 Report
This study is well written and pleasant to read. Rigorous methodology and beautiful images are provided. The aim is to assess the quality of sealer penetration into the tubules and the adaptation to the dentinal wall of a novel Bio-ceramic Calcium-Silicate sealer, both after classic syringe irrigation and after the use of Endo-Activator.
Specific comments:
In the author affiliations: the same city is spelled differently in1) compared to 3)
Line 25: The use of the word “percentage” seems confusing here. It could mean a lot of different things. Wouldn’t “circumferential percentage” be more appropriate? (also applicable for further use of the term “percentage”)
Line 46: why is there no © after Self Adjusting File?
Line 57: “new experimental novel” Sounds redundant. Choose either “new experimental” or “novel”.
Line 77: Tooth collection: did patients provide an informed consent after receiving explanations about this study? What were the reasons for extraction of these teeth?
Line 105; Were gutta cones fitted or calibrated?
Line 227: What does NE mean? Shouldn’t this be CN?
Line 243: “physical proprieties and possible changes in surface tension of the dentine”
Do you have any reference for this? I don’t find anything on surface tension in [59]. I find it hazardous to speculate that Endo-Activator could change the surface tension of dentine.
Line 271 and 272: “Guy LE Brun”, should be “Guy Le Brun” according to the title page.
General comment:
It is unfortunate that there is no comparison with other activation techniques and no comparison with other cements.
Author Response
Dear reviewer,
We would like to thank you for your valuable comments that we took into consideration. We are very pleased to know that you appreciate our work and recognize its interest and quality.
You can find below a detailed response to all the points.
Best regards,
The authors
Point #1: In the author affiliations: the same city is spelled differently in1) compared to 3)
Reply to point #1: In Line 12, we corrected the spelling and replaced Riad El Solh by Riad el-Solh
Point #2: Line 25: The use of the word “percentage” seems confusing here. It could mean a lot of different things. Wouldn’t “circumferential percentage” be more appropriate? (also applicable for further use of the term “percentage”)
Reply to point #2: In Lines 25, 27, 217, 224, 225, Table 1, 254, 258,264, we considered this remark and replaced the word “percentage” by “circumferential percentage” in the manuscript.
Point #3: Line 46: why is there no © after Self Adjusting File?
Reply to point #3: In Line 46, we added ®: Adjusting File®.
Point #4: Line 57: “new experimental novel” Sounds redundant. Choose either “new experimental” or “novel”.
Reply to point #4: In Line 72, we deleted new experimental and replaced the sentence by the following one: A novel tricalcium silicate (NTS)-based sealer composed of tricalcium silicate, tantalum, and calcium oxide has been recently developed [13].
Point #5: Line 77: Tooth collection: did patients provide an informed consent after receiving explanations about this study? What were the reasons for extraction of these teeth?
Reply to point #5:
- Since our study does not contain any research involving human participants or animals, formal consent of patients is not required.
- Extraction of these teeth was not in correlation with this study. They were collected from several dental clinics.
Point #6: Line 105: Were gutta cones fitted or calibrated?
Reply to point #6: During the obturation, we used the gutta-percha cone (ProTaper Universal F4, Dentsply Maillefer) calibrated that fits the ProTaper® Finisher F4. Same technique was previously used in other studies published in the literature. See for instance:
1. McMichael, G.E.; Primus, C.M.; Opperman, L.A. Dentinal tubule penetration of tricalcium silicate sealers. J. Endod. 2016, 42, 632–636, doi: 10.1016/j.joen.2015.12.012.
2. Jeong, J.W.; DeGraft-Johnson, A.; Dorn, S.O.; Di Fiore, P.M. Dentinal Tubule Penetration of a Calcium Silicate-based Root Canal Sealer with Different Obturation Methods. J. Endod. 2017, 43, 633–637, doi: 10.1016/j.joen.2016.11.023.
3. Aksel, H.; Küçükkaya Eren, S.; Puralı, N.; Serper, A.; Azim, A.A. Efficacy of different irrigant prorocols and application systems on sealer penetration using a stepwise CLSM analysis. Microsc. Res. Tech. 2017, 80, 1323–1327, doi:10.1002/jemt.22944.
Point #7: Line 227: What does NE mean? Shouldn’t this be CN?
Reply to point #7: In Line 319, we do agree with the reviewer’s comment. we replaced NE by CN.
Point #8: Line 243: “physical proprieties and possible changes in surface tension of the dentine”
Do you have any reference for this? I don’t find anything on surface tension in [59]. I find it hazardous to speculate that Endo-Activator could change the surface tension of dentine.
Reply to point #8: The reviewer has a good point. It was a speculation from the authors that Endo-Activator could change the surface tension of dentine. The reference 59 was for the next sentence: “Additionally, a possible explanation for this could be that this irrigant activation unobstructs dentinal tubules and exposes dentine collagen fibrils. As a result, sealer penetration into dentinal tubules could be improved [59]”
If you advise we can delete it.
Point #9: Line 271 and 272: “Guy LE Brun”, should be “Guy Le Brun” according to the title page.
Reply to point #9: In Line 420, we replaced LE by Le.
Point #10: It is unfortunate that there is no comparison with other activation techniques and no comparison with other cements
Reply to point #10: As pointed out by the reviewer, the aim of our work is to assess the quality of sealer penetration into the tubules and the adaptation to the dentinal wall of a novel Bio-ceramic Calcium-Silicate sealer, both after classic syringe irrigation and after the use of Endo-Activator. Comparison with other cements and other activation techniques are very interesting ideas; They definitely deserve to be addressed in other studies.

Round 2
Reviewer 3 Report
225 Circumferential should not take a Capital letter : should be "circumferential" in this case
305 most requirement: should be “one of the most important requirements” or “one of the major requirements”
Author Response
Dear reviewer,
We would like to thank you for the comments that we took into consideration.
You can find below the changes that we made.
Best regards,
The authors
-Revised text in Line 5, we deleted the capital letter « C » and we replaced Cirunferential by circumferential
-Revised text in Line 305, we corrected and replaced most requirement by one of the most important requirements